# OpenReview forum: "Inverse Scaling in Test-Time Compute"
_TMLR — Accepted by TMLR_

### Review · Reviewer_6su3 · 2025-08-14

**Summary Of Contributions:**

The authors investigate reasoning length as a failure mode in frontier Large Reasoning Models (LRMs).
The authors contributions are:
i) Designing the evaluation pipeline, together with a benchmark dataset of tasks, spanning four different categories.
ii) The authors find interesting failure behavior in long chains of reasoning, e.g. models getting distracted by excessive information, or moving from having good priors to attaining to spurious correlations.
Overall, this is an empirical work evaluating frontier LRMs and the contributions are mostly the experimental design choices and the insights derived from it.

**Audience:**

Yes

**Audience Explanation:**

As LRMs become more popular and companies invest heavily in test-time compute, understanding when these systems fail is crucial. This work is valuable because it finds surprising cases where using more compute actually makes performance worse, challenging the common belief that more reasoning always helps. What makes this more important is that the authors could run large, expensive experiments that most researchers can't afford, giving us rare insights into how these advanced models really work. Since companies like Anthropic and OpenAI share little about their private models, these results fill an important knowledge gap for the broader research community.

**Broader Impact Concerns:**

The authors have a Limitations section that addresses some of this work's limitations. I don't think it has any major _ethical_ concerns, since the tasks are synthetic and no broader statements are made.

**Claims And Evidence:**

Yes

**Claims Explanation:**

This is mostly an empirical work, so the claims have to be supported by the experimental findings, which is certainly true. Concretely, the authors list 5 findings in the abstract and introduction that are supported in the results section(s). In particular, I'd like to highlight that the authors are precise in the statement "we show cases where longer reasoning deteriorates performance...". This is important because scaling benefits or damages are always relative to the target measure, i.e., an specific task (or capability). Although this is addressed in the paper, and therefore my answer is 'yes', I clarify in "Requested Changes" how this can be further improved.

**Requested Changes:**

The paper is overall well-written and as I mentioned previously, the claims are supported by empirical evidence. Still, there are a few things I think could be done to further improve the manuscript:

- The fact that test-time compute works, or isn't harmful, to the inverse-scaling challenge tasks (built in 2023) isn't sufficient to say that necessarily the scaling factors differ from train and test (first two paragraphs of Sec 4). This is quite a broad statement and I don't think we have sufficient evidence to believe that (although it could certainly be true). Since these models are black boxes (we have no idea how or on what they were trained), assessing them with known benchmark is quite tricky.

- The tasks are quite "synthetic", in the sense that they are borderline adversarial, e.g. inserting random probability statements in an arithmetic task. I wonder if the authors ask the LLM to map the questions into a coherent story (without removing information), and ask the question, the performance goes up. This is a known issue with synthetic/adversarial evaluations of LLMs, unless we're modeling adversarial users, this is quite unlikely to happen in real life. I wonder if the authors could do a small experiment (maybe on the arithmetic task), transforming the inputs.

- I think Takeaway 2 is the most important, "training may have incentivized recognition of known problems and algorithms over correct reasoning." This is actually a core problem with private models. We have no idea whether they are just being trained to recognize known problems. And, if they are, what's the extent of the harm? This could be more discussed and highlighted, perhaps in the intro.

- Takeaway 3 is showcasing that test-time compute is being used to improve prediction, and this can cause the use of spurious correlations. This is actually a well-known fact in causal inference literature,  if you don't specify a causal model, spurious correlations are used as a shortest path to prediction. Maybe the authors can test whether injecting a prior knowledge in the prompt is sufficient to keep the model attaining to it, or if even then, it runs off track.

- Minors: some plots are very hard to read, especially in the appendix, as the lines collapse. Also, I think we're more interested in comparing if different models have the same trend or not, rather than across "# of examples", "# of distractors", etc.

---

> ### Author Response · Authors · 2025-10-27
> **Response to Reviewer 6su3 [1]**
>
> Thank you for your constructive feedback. We're pleased that you find our work valuable and agree that our claims are well-supported. We address each point below:
>
> > The fact that test-time compute works, or isn't harmful, to the inverse-scaling challenge tasks (built in 2023) isn't sufficient to say that necessarily the scaling factors differ from train and test (first two paragraphs of Sec 4).
>
> Thank you for spotting this. We agree that our statement about train vs test-time scaling differences is too broad, given the existing evidence. The sentence was added to motivate the need for studying test-time scaling behaviors. Therefore, we revised the last sentence of the first paragraph of Section 4 to:
>
> *“The absence of test-time inverse scaling in existing benchmarks---including those designed to reveal training-time inverse scaling---motivates our development of new evaluation tasks specifically targeting potential test-time compute failure modes.”*
>
> > Synthetic task naturalness
>
> We appreciate the reviewer's concern about the synthetic nature of our tasks. To address this, we conducted an additional experiment using a more naturalistic dataset construction method while maintaining our research question: *do models fail to filter salient but irrelevant information under increased reasoning effort?*
>
> Rather than using template-based distractor insertion, we employed an LLM (Claude Sonnet 4.5) to generate natural-sounding distractors. Specifically, we prompted the model to use extended thinking with a 16,000-token thinking budget to generate natural-sounding distractors and re-evaluate whether the distractors are plausible and do not change the final answer. The model was given simple background-question pairs as templates similar to the original task (e.g., "You have an apple and an orange" and "Calculate how many fruits do you have") and instructed to introduce five plausible, contextually-relevant distractors that appear informative but do not affect the answer. This process yielded 200 instances across 10 question templates, with distractors phrased in fluent natural language rather than template insertions.
>
> We maintained using salient-but-irrelevant distractors rather than coherent narratives. This design choice addresses our research question which focuses on understanding the effect of salient distractors during extended reasoning. Creating fully coherent stories would alter the task, where all details become relevant by construction, especially for trivial counting problems where the answer is directly inferrable. This would eliminate the distractor effect we aim to study. To add, our approach captures important aspects of real-world scenarios (e.g., users frequently include information they believe relevant that isn't, RAG systems return tangentially related documents, and long prompts often mix relevant and irrelevant content). The apparent "adversarial" nature stems from the deliberate contrast between trivial correct answers and plausible distractors. This contrast allows us to isolate and study the failure mode while minimizing the confounding factors.
>
> In this more naturalistic setup, we find that inverse scaling trends persist: Claude Sonnet 3.7 and Sonnet 4 show performance degradation with increased reasoning budgets, demonstrating that the phenomenon is not merely an artifact of awkward synthetic language but reflects a genuine challenge in maintaining focus on task-relevant information when presented with fluent, plausible distractors.
>
> Full details of this experiment are provided in Appendix G.1.
>
> > Takeaway 2 emphasis in Section 1
>
> We agree that this is an important finding. We revised a sentence in the introduction section to explicitly highlight this finding:
>
> *“In simple counting tasks with distractors (Section 4.1), Claude models become increasingly distracted by irrelevant information as they reason longer, while OpenAI o-series models resist distractors but show tendency to apply memorized solution patterns when recognizing familiar problem framings rather than genuinely reasoning through the actual question.”*

---

> ### Author Response · Authors · 2025-10-27
> **Response to Reviewer 6su3 [2]**
>
> > Spurious correlation mitigation
>
> We appreciate the reviewer's suggestion to test whether providing prior knowledge helps models avoid spurious correlations.
>
> Using the Grades Regression task, we tested whether prior knowledge could guide models toward correct feature usage. We sampled 100 examples from the zero-shot condition and appended prior statements to each prompt (three variations of priors, resulting in 300 total examples). We tested two variants: explicit priors that directly mention study hours as the dominant predictor (e.g., "Recent educational studies suggest that study hours is often the strongest predictor of academic outcomes"), and implicit priors that suggest dominance without specifying which feature (e.g., "Recent educational studies suggest that time investment is often the strongest predictor of academic outcomes").
>
> Both explicit and implicit priors led to inverse scaling across all tested models. Analysis of reasoning traces shows that models fixate on the prior statements and over-rely on a single feature (study hours) while neglecting other relevant features. While study hours is indeed the most correlated feature, the underlying data shows that other factors (e.g., stress level) also affect a student’s grade.
>
> This contrasts with the in-context learning results presented in Section 4.2, where in-context examples help models maintain balanced consideration of multiple features. This suggests that concrete examples are more effective than statements of prior knowledge in preventing reliance on spurious correlation.
>
> Full details are provided in Appendix G.2.
>
> > Plot improvements
>
> Thank you for spotting this, we improved the readability of the figures.

---

### Review · Reviewer_5ArC · 2025-09-03

**Summary Of Contributions:**

The paper proposes evaluation tasks for Large Reasoning Models that span four categories: counting tasks with distractors, regression tasks with spurious features, deduction tasks with constraints, and advanced AI risks. The experiments investigate how different models respond to these distractors and how the performance on the tasks is impacted. The work investigates whether there are scenarios where longer reasoning (an increased reasoning budget) leads to poorer performance.

STRENGTHS

S1. The paper investigates a very interesting aspect of the reasoning behaviour of LRMs and constructs instructive problems to explore performance deficiencies.

S2. The experiments are extensive, with exploration across numerous LRMs and problem settings. The experiments are carefully conducted with conclusions drawn based on confidence intervals.

S3. The paper is clearly written; the overall hypothesis and claims are evident, and the core takeaways are specified.

WEAKNESSES

W1. The controlled experiments are conducted with a single prompt strategy for controlling/affecting the test-time compute that should be allocated to the presented problem. In some places, the paper errs on the side of making more general conclusions about all strategies for scaling test-time compute.

W2. The prompt strategy appears to have effects on test-time compute that differ dramatically from model to model and from one task to another. This can make it more challenging to draw clear conclusions from the results.

W3. The presentation of some plots should be improved.

**Audience:**

Yes

**Audience Explanation:**

The topic is of significant interest to a relatively large group of researchers in the TMLR community and the experiments, results, and analysis are illuminating.

**Claims And Evidence:**

Yes

**Claims Explanation:**

The paper makes the following major claims:

(1) the paper shows cases where longer reasoning leads to a deterioration in performance, demonstrating an inverse scaling relationship between test-time compute and accuracy;

(2) the experiments show that longer reasoning processes may amplify flawed heuristics;

(3) the findings reveal a critical gap between short and extended reasoning alignment;

(4) Extended reasoning introduces safety risks; scaling up test-time compute may amplify model-specific concerning behaviours.

The claims are, for the most part, supported by accurate, convincing and clear evidence. In support of the first claim above, the paper “shows cases”; while the observed deterioration in performance may not always occur, and while it may be dependent on the applied prompts as much as the constructed problems, the experiments do demonstrate multiple examples of this behaviour.

One concern is that the first takeaway states “Scaling up test-time compute reduces the accuracy of most models on simple counting tasks with distracting information, particularly in the natural overthinking setup.” The paper only provides evidence that scaling up test-time compute in the specific way adopted in the experiments leads to reduced accuracy. There are many different ways to scale up test-time compute, and it is possible that some of the other strategies proposed in the literature do not exhibit the same effect.

**Requested Changes:**

RQ1. Please add some commentary about the sensitivity to the form of the prompt and/or details about any experiments conducted in the process of designing/tuning the prompt (i.e., was this the first thing that was tried or were other approaches explored first?). Do the observations remain consistent if different mechanisms are used for triggering different degrees of thinking? Given the extent of the experimental work, I don’t think there is a need for additional experiments exploring the impact of changing the prompt, but the paper would be strengthened by some discussion of this topic.

RQ2. Could the authors please add text (probably in Appendix A) to explain how the confidence intervals are estimated and clarify that any employed assumptions (e.g., approximate normality) are clearly met by the data.

RQ3. The labelling of the x-axes in Figures 3-6 and 8 makes comparison very difficult. Please strive for more consistent labelling that provided better coverage of the axis. For example, the top-left panel of Figure 3 is particularly bad, with no marks at all for the right half of the plot. DeepSeek in Figure 8 has no labels at all.

RQ4. The comparison in Figure 3 with the “controlled’ resulting in a range of at most ~3K tokens for Claude Opus 4, ~600 tokens for o3, but ~12K tokens for DeepSeek R1. This doesn’t seem particularly “controlled”, and it raises concerns that the structure of the prompt (which according to Appendix A is inspired by the Claude Code documentation) is having significantly different effects across the models. It would be helpful if the authors could add some commentary to explain the significant disparity in token count and the extent of the achieved token increase.

---

> ### Author Response · Authors · 2025-10-27
> **Response to Reviewer 5ArC [1]**
>
> Thank you for your constructive feedback. We're pleased that you find our work valuable and agree that our claims are well-supported. We address each point below:
>
> > RQ1. Please add some commentary about the sensitivity to the form of the prompt and/or details about any experiments conducted in the process of designing/tuning the prompt (i.e., was this the first thing that was tried or were other approaches explored first?). Do the observations remain consistent if different mechanisms are used for triggering different degrees of thinking? Given the extent of the experimental work, I don’t think there is a need for additional experiments exploring the impact of changing the prompt, but the paper would be strengthened by some discussion of this topic.
>
> Thank you for this suggestion. To clarify, we decide to use the current prompt (i.e., keywords combined with token budgets) because it successfully induces varying reasoning lengths across all models. While we did not extensively explore alternative prompting strategies in the controlled setup, our experimental design addresses prompt sensitivity through complementary approaches:
>
> - The natural overthinking setup removes explicit budget instructions entirely, yet shows similar inverse scaling patterns (and stronger in the Constraint Satisfaction Problem task);
> - The cautioned overthinking setup (Appendix D) tells models they need not exhaust the provided reasoning budget. The consistency of results across these different prompting mechanisms suggests our findings are robust across prompt designs.
>
> We added a discussion of our prompt design to Appendix A:
>
> *As shown in Figure 2 and Figure 11, models interpret these controlled overthinking prompts with varying degrees of compliance---DeepSeek R1 generates significantly more tokens than Claude or OpenAI models given an identical prompt. Nonetheless, we select this approach and do not extensively tune the prompt as it successfully induced increasing reasoning lengths across all models. To address potential sensitivity to prompt phrasing, we also evaluate a natural overthinking variant where we omit any keywords that instruct the model to think for longer. [...]
> Additionally, we evaluate a cautioned overthinking variant where we explicitly tell the models that they need not exhaust the budget (see Appendix D), which also results in inverse scaling in the evaluation tasks. The consistency of inverse scaling patterns across these different prompting approaches suggests that our findings are robust to variations in prompts.*
>
> > RQ2. Could the authors please add text (probably in Appendix A) to explain how the confidence intervals are estimated and clarify that any employed assumptions (e.g., approximate normality) are clearly met by the data.
>
> Thank you for asking for more details. Confidence intervals are estimated using the standard error of the mean (SEM) with a 95% confidence level (z-score = 1.96). More precisely:
> For accuracy metrics, within each group, we compute the mean and standard deviation of accuracy values across all questions in that bucket, then calculate SEM = std / sqrt(n), where n is the number of questions. The 95% confidence interval is ±1.96 × SEM.
> For RMSE metrics, we first convert squared errors to RMSE values (by taking the square root) for each individual question, then compute the mean and SEM of these RMSE values within the bucket. We report ±SEM as error bars.
> For x-axis (cost) error bars: We calculate the SEM of costs across all questions within each bucket.
> Regarding statistical assumptions: Each bucket contains results from all questions in the benchmark (typically thousands of questions, e.g., 1500 for Misleading Python in the controlled overthinking setup (500 questions across 3 random seeds)). With sample sizes in the thousands, the sampling distribution of the mean is approximately normal, regardless of the underlying distribution of individual question responses. This justifies our use of normal-theory confidence intervals.
>
> > RQ3. The labelling of the x-axes in Figures 3-6 and 8 makes comparison very difficult. Please strive for more consistent labelling that provides better coverage of the axis.
>
> We appreciate your attention to figure readability. We have standardized the x-axis labeling across figures (in the main text and appendix).

---

> ### Author Response · Authors · 2025-10-27
> **Response to Reviewer 5ArC [2]**
>
> > RQ4. The comparison in Figure 3 with the “controlled’ resulting in a range of at most ~3K tokens for Claude Opus 4, ~600 tokens for o3, but ~12K tokens for DeepSeek R1. This doesn’t seem particularly “controlled”, and it raises concerns that the structure of the prompt (which according to Appendix A is inspired by the Claude Code documentation) is having significantly different effects across the models. It would be helpful if the authors could add some commentary to explain the significant disparity in token count and the extent of the achieved token increase.
>
> By "controlled overthinking," we mean explicitly prompting models to think longer, not necessarily achieving identical token counts across models. In general, models generate vastly different token amounts from identical prompts. In our results, all show inverse scaling patterns. We plot performance against actual tokens generated rather than requested budgets specifically to account for this variation. The key finding is that performance degrades as reasoning length increases, regardless of whether a model produces hundreds or thousands of tokens.
>
> We added a commentary in Section 3, noting this disparity:
>
> *“Models generate different token counts from identical \textit{controlled overthinking} prompts, but all models generate more tokens as we increase the requested reasoning budget, which is sufficient to induce the cases of overthinking central to our study.”*

---

### Review · Reviewer_Ex9i · 2025-10-14

**Summary Of Contributions:**

The paper introduces novel evaluation tasks designed to reveal inverse scaling trends in test-time compute for large reasoning models, where increased reasoning length (i.e., more tokens generated during inference) leads to degraded performance, contrary to typical positive scaling observations. Key contributions include: (1) constructing three main task categories — simple counting with distractors (e.g., misleading math/Python snippets), regression with spurious features (e.g., predicting student grades from XML records with irrelevant data), and deduction with constraint tracking (e.g., Zebra puzzles), plus evaluations on model-written tasks for AI risks; (2) identifying five distinct failure modes across models, such as distraction by irrelevancies in Claude models, overfitting to framings in OpenAI o-series, shifting to spurious correlations, loss of focus in deduction, and amplified concerning behaviors like self-preservation; (3) experimental setups comparing controlled, natural, and cautioned overthinking to isolate effects; and (4) qualitative analyses showing how extended reasoning amplifies flawed heuristics rather than correcting them. The work highlights gaps in current benchmarks and implications for AI alignment.

Key strengths:

- Innovative task design that uncovers underexplored failure modes in test-time scaling
- rigorous experiments across frontier models (e.g., Claude series, o-series, open-weight like DeepSeek R1)
- clear visualizations and qualitative insights
- relevance to safety and alignment research.

Key weaknesses:

- Reliance on synthetic tasks may limit generalizability to real-world scenarios
- limited analysis of why certain models (e.g., o3) show more stability

**Audience:**

Yes

**Audience Explanation:**

This topic is of high relevance to the TMLR community, particularly those studying reasoning, test-time scaling, and alignment of LLMs. The findings highlight an important counterintuitive behavior -- performance degradation with increased inference compute -- which has implications for both practical design and scientific understanding of reasoning in LLMs.

**Broader Impact Concerns:**

The paper raises valid concerns about amplified misaligned behaviors (e.g., self-preservation, which could relate to deception or power-seeking in deployed systems). No major ethical concerns arise from the work itself.

**Claims And Evidence:**

Yes

**Claims Explanation:**

The empirical evidence is extensive and convincingly supports the claims. The paper uses multiple benchmarks, ablation analyses, and cross-model comparisons to demonstrate the phenomenon of inverse scaling. More theoretical grounding could strengthen the work.

**Requested Changes:**

1. Strengthen causal claims on reasoning length vs. performance degradation: The current controlled overthinking setup relies heavily on prompts like "think harder" or explicit budgets, which may confound length with induced overcomplication (e.g., Figures 3-4 show drops potentially tied to phrasing rather than compute). This risks equating chain growth with "overthinking" without proving causality, weakening the core inverse scaling argument. It would be great if authors could add experiments isolating reasoning length (e.g., via token truncation or iterative refinement) from prompting effects, and analyze token quality (e.g., repetition metrics) to rule out degradation from verbose but unhelpful outputs.
2. It would be great if authors could discuss if task generation (e.g., synthetic distractors) inadvertently favors certain failure modes, and test with human-curated distractors. This would boost the real-world applicability, especially since standard benchmarks like GSM8K show no inverse scaling, hinting that your effects might just be quirks of these custom traps, not a broad LRM issue.

---

> ### Author Response · Authors · 2025-10-27
> **Response to Reviewer Ex9i**
>
> Thank you for your constructive feedback. We're pleased that you find our work valuable and agree that our claims are well-supported. We address each point below:
>
> > limited analysis of why certain models (e.g., o3) show more stability
>
> We acknowledge this as a limitation of our work. Since the models we evaluated have undisclosed pretraining methods, we cannot determine the factors that are responsible for their stabilities or instabilities. Our contribution focuses on documenting which failure patterns occur across different models.
>
> > Strengthen causal claims on reasoning length vs. performance degradation: The current controlled overthinking setup relies heavily on prompts like "think harder" or explicit budgets, which may confound length with induced overcomplication (e.g., Figures 3-4 show drops potentially tied to phrasing rather than compute). This risks equating chain growth with "overthinking" without proving causality, weakening the core inverse scaling argument. It would be great if authors could add experiments isolating reasoning length (e.g., via token truncation or iterative refinement) from prompting effects, and analyze token quality (e.g., repetition metrics) to rule out degradation from verbose but unhelpful outputs.
>
> We appreciate the reviewer's concern about establishing causality between reasoning length and performance degradation, with prompting being a potential confounder. We designed the Natural Overthinking setup to address this concern. In this setup, models receive no explicit instructions like "think harder" or token budgets—we simply instruct the model to think. This eliminates potential confounding from prompting effects. We observe that within each model, longer reasoning traces correlate with performance degradation on our tasks. This provides evidence that the phenomenon occurs under natural conditions, not only when explicitly prompted to think harder.
>
> Regarding additional analysis on the reasoning quality, we conducted qualitative analysis of reasoning traces for models where they are accessible (i.e., Claude models and open-weight models). We do not observe token repetition or other signs of output quality degradation. Models occasionally exhibit circular reasoning patterns, but the reasoning traces themselves remain coherent and do not show quality degradation (e.g., repetitive tokens or nonsensical text). The performance drops appear to stem from over-analyzing the question rather than from generating low-quality tokens. For OpenAI models (o1, o3-mini), reasoning traces are not currently disclosed, preventing similar analysis. We describe the observed failure modes in detail in Section 4.1, 4.2, 4.3, and 5. We also provide the qualitative examples in Appendix I.
>
> > It would be great if authors could discuss if task generation (e.g., synthetic distractors) inadvertently favors certain failure modes, and test with human-curated distractors. This would boost the real-world applicability, especially since standard benchmarks like GSM8K show no inverse scaling, hinting that your effects might just be quirks of these custom traps, not a broad LRM issue.
>
> We appreciate the reviewer's point about task construction potentially favoring specific failure modes. We argue that discovering these "quirks" is valuable because they reveal model brittleness.
>
> Even if the failure modes are specific to certain task constructions, they demonstrate that models can be fragile to simple distracting framings. Standard benchmarks like GSM8K may not exhibit inverse scaling because they lack these distractors by design, not because models are robust to them. Our contribution is identifying conditions under which longer reasoning can lead to worse accuracy, which helps understand failure modes of the current reasoning models.  Such simple distractors can naturally occur in real-world scenarios: users frequently include context they believe is relevant but isn't, retrieval systems (e.g., RAG) return tangentially related documents, and automated pipelines aggregate information with varying relevance.
>
> To add, in response to Reviewer 6su3, we add a Misleading Math experiment with LLM-generated distractors that are more realistic (Appendix G.1). The results show that inverse scaling persists even with more naturalistic construction, suggesting the phenomenon extends beyond synthetic template-based distractors.

---

### Author Response · Authors · 2025-10-27
**Global Response**

We thank all reviewers for their time and constructive comments. We appreciate their recognition of the convincing empirical evidence supporting our claims (*6su3, 5ArC, Ex9i*), of the high relevance and importance of the research topic to the TMLR community (*6su3, 5ArC, Ex9i*), of the rigorous experimental design across frontier models (*5ArC, Ex9i*), of the clear presentation and well-defined takeaways (*5ArC, Ex9i*), and of the counterintuitive and important findings of inverse scaling in test-time compute (*6su3, Ex9i*). In response to their feedback, we have made several amendments to the manuscript:

1. **Softened claims about train vs test-time scaling (Section 4)**: Revised the opening of Section 4 to avoid overstating conclusions from existing benchmarks about differences between training and test-time scaling factors.
2. **Added LLM-generated distractor experiment (Appendix G.1)**: Conducted a new experiment using Claude Sonnet 4.5 to generate natural-sounding distractors rather than template-based ones. Results show inverse scaling persists with more naturalistic task construction.
3. **Added prior knowledge experiment (Appendix G.2)**: Tested whether providing explicit or implicit prior statements helps models avoid spurious correlations in the Grades Regression task. Found that priors lead to inverse scaling, contrasting with the success of in-context examples.
4. **Enhanced Introduction**: Revised text to explicitly highlight the finding about the pattern-matching tendency in the OpenAI o-series models.
5. **Added prompt design discussion (Appendix A)**: Clarified our prompt selection rationale and documented robustness across different prompting approaches (controlled, natural, and cautioned overthinking variants).
6. **Added statistical methodology details (Appendix A)**: Documented confidence interval estimation methods (SEM with 95% CI) and justified the use of normal-theory intervals given large sample sizes.
7. **Added commentary on token count variation (Section 3)**: Explained that models generate vastly different token counts from identical prompts, but all show inverse scaling patterns regardless of absolute token counts.
8. **Improved figure readability**: Standardized x-axis labeling across all figures and enhanced visual clarity.
9. **Additional related works**: Added several relevant papers discovered after the initial submission.

All changes are marked in blue font in the updated version.

---

### Decision · Action_Editor_Tikk · 2025-12-06

**Recommendation:** Accept as is

**Audience:**

Yes

**Audience Explanation:**

All reviewers concur this is a very timely paper and shed light on a phenomenon that plagues frontier models thus shedding light on possible future fixes for this problem.

**Claims And Evidence:**

Yes

**Claims Explanation:**

High quality paper with extensive experiments showing that thinking longer might actually decrease performance (inverse scaling) due to different reasons such as amplification of irrelevant information in the context and reinforcement of problematic thinking patterns. Multiple benchmarks, ablation analyses and especially multiple models (o3, claude, deepseek) are used to demonstrate the interesting phenomenon.